# An Overview of the Relevance of Human Gut and Skin Microbiome in Disease: The Influence on Atopic Dermatitis

Maria Pia Ferraz [1,2,3]

1   Departamento de Engenharia Metalúrgica e de Materiais, Faculdade de Engenharia da Universidade do Porto, 4200-465 Porto, Portugal; mpferraz@ineb.up.pt
2   i3S—Instituto de Investigação e Inovação em Saúde, Universidade do Porto, 4200-135 Porto, Portugal
3   INEB—Instituto de Engenharia Biomédica, Universidade do Porto, 4200-135 Porto, Portugal

**Abstract:** It is acknowledged that humans have a diverse and abundant microbial community known as the human microbiome. Nevertheless, our comprehension of the numerous functions these microorganisms have in human health is still in its early stages. Microorganisms belonging to the human microbiome typically coexist with their host, but in certain situations, they can lead to diseases. They are found in several areas of the human body in healthy individuals. The microbiome is highly diverse, and its composition varies depending on the body site. It primarily comprises bacteria that are crucial for upholding a state of well-being and equilibrium. The microbiome's influence on atopic dermatitis development was, therefore, analyzed. The importance of maintaining a balanced and functional commensal microbiota, as well as the use of prebiotics and probiotics in the prevention and treatment of atopic dermatitis were also explored. The skin microbiome's association with atopic dermatitis will allow for a better understanding of pathogenesis and also exploring new therapeutic approaches, making the skin microbiome an increasingly relevant therapeutic target.

**Keywords:** human microbiome; skin microbiota; atopic dermatitis; *S. aureus*; *S. epidermidis*

## 1. Introduction

The human body is colonized by trillions of microorganisms, including distinct species of bacteria, fungi, viruses, and protozoa. Collectively, these microorganisms and their genetic makeup form a constantly changing microbial community that resides in various regions of the body and significantly influences the well-being, illness, maturation, and progression of their host, being commonly known as the human microbiome [1].

The microbiome is the genetic heritage of microorganisms that live with humans, classified as the community of microscopic organisms that colonize various areas of the human body, whether superficial or deep, such as the genitourinary system, respiratory system, and gastrointestinal system, among others. Its development is a dynamic process that varies throughout life, coexisting with its host and actively participating in many biological processes in the human body [1]. This involvement of the microbiome in various functions makes it essential for maintaining homeostatic balance, and its impairment can lead to problems for the host, such as the emergence of several diseases, including cardiovascular diseases [2]; autoimmune diseases, namely in rheumatoid arthritis, Type 1 diabetes, atopic asthma, and atopic dermatitis [3]; and colon cancer [4].

Contrary to the human genome, which is rarely influenced by external factors, the human microbiome is characterized by some volatility, as demonstrated by the use of antibiotics, changes in diet, or states of infection in the host organism, which can subsequently lead to significant modifications in host microorganisms. Lifestyle and exercise also have a powerful impact on the composition of the human microbiome [5–7]. Therefore, several factors shape the microbiome, preventing the development and prevalence of various diseases and leading to a healthier life. Among these factors, the Mediterranean diet, medicinal

therapies, the use of probiotics and prebiotics, and even fecal bacterial transplantation stand out [8].

The human body site that harbors the largest number and diversity of microorganisms is the gastrointestinal tract, specifically the intestine, which exerts a greater influence on human homeostatic mechanisms [9]. Several diseases and their risk factors have been identified as examples of the interaction between the host and the gut microbiome [10,11]. With the development of culture-independent methods, it has become possible to better identify microorganisms and their genomes. These microorganisms can have either harmful or beneficial effects on their host. Concerning skin microbiota, *Staphylococci* species are considered harmful, *Cutibacterium* often being cited as beneficial [12]. Concerning gut microbiota, the presence of *Lactobacillus* and *Bifidobacterium* species is usually considered beneficial [13]. There are indications that microorganisms, both on the skin and in the gut, might impact the progression of atopic dermatitis. While antiseptic treatments have been in use for many years, recent advancements in traditional-culture-based techniques and cutting-edge metagenomics are shedding light on the potential of targeted dysbiosis treatment as a potential component of an integrated therapeutic approach in the future [14–16].

Therefore, this manuscript aimed to provide an updated overview of the intriguing relationship between the human microbiome and diseases, seeking to establish the importance of the microbiome in health and disease, focusing on the case of atopic dermatitis. For this purpose, a literature search was conducted using the keywords human microbiome, skin diseases, atopic dermatitis, probiotics, and prebiotics, in a scientific database, selecting the most-relevant scientific articles for this work concerning the purpose of the manuscript.

## 2. Human Microbiome

In a healthy adult, the human body usually contains around ten-times as many microbial cells as it does human cells, primarily because of the extensive variety of microorganisms found in the gastrointestinal tract. The microorganisms that constitute the human microbiome can coexist in relationships of commensalism, where the association between the microorganism and the host is seemingly neutral, without identifiable benefits or harm, or in relationships of mutualism, where the interaction between both is beneficial, although their survival does not require them living together [1].

The development of the human microbiome is a dynamic process involving different stages and exhibits notable differences in diversity and variety. The microbial distribution across various sites of the human body, including the oral cavity, skin, gastrointestinal tract, and genitourinary tract, has an impact on human health. The vast diversity of and variety within the human microbiome can be observed between different individuals or even within a single individual [17,18]. This is because, due to its dynamism, the microbiome differs from person to person and even among different anatomical locations of the same individual. Each body part is characterized by a unique ecological community of microorganisms, making each microbiome distinct [17,18]. These interpersonal differences are influenced by genetic background, geographic origin, age, lifestyle choices, diet, premature microbial exposure, as well as the regularity of antibiotic or probiotic use [1].

The human microbiome provides a fundamental internal ecosystem for numerous physiological processes, among which some can be highlighted, such as protection against pathogens, nutrient processing, stimulation of the angiogenesis, development, and maintenance of the intestinal epithelial barrier, and the maintenance of the immune system, among others. It evolves in conjunction with the host to form a superorganism, where both the microbiome and the host interact and rely on each other for optimal functioning [8]. Therefore, characterizing and understanding the complexity of the microbiome can serve as a diagnostic tool, creating opportunities to improve the quality of life through microbiome manipulation.

It was in line with this thinking that the Human Microbiome Project (HMP) was created, an international program aimed at generating research foundations that allow for comprehensive characterization of the microbiome and analysis of its role in health and

disease. The project seeks to understand all the microorganisms existing in our bodies and acquire knowledge about their composition in healthy individuals, as well as how it is altered in pathological conditions. This understanding can contribute to the prevention and treatment of diseases [19]. As a consequence, several research groups have accomplished characterizing the composition of the human microbiome as being dominated by four phyla: Actinobacteria (including *Bifidobacterium* and *Corynebacterium* genera), Bacteroidetes (genus *Bacteroides*), Firmicutes (which include *Bacillus*, *Lactobacillus*, *Staphylococcus*, and *Streptococcus* genera), and Proteobacteria (including *Escherichia*, *Salmonella*, *Vibrio*, *Campylobacter*, and *Helicobacter* genera) [19–21].

### 2.1. Microbiome Types
#### 2.1.1. Oral Cavity

The oral cavity is considered one of the most-dynamic ecosystems in the human body, housing approximately 50 to 100 billion bacteria, along with other microorganisms such as fungi and viruses, totaling around 700 identified bacterial species. This ecosystem comprises various structures, including teeth, lips, and gums, among others, and each of them serves as an ecological niche that promotes the growth of microorganisms. As a result, different environments are created within the oral cavity, leading to the development of distinct microbiomes [22]. Factors such as the consumption of sugars and amino acids and the constant production of saliva contribute to this variation. While most of these bacteria are commensal species, depending on the individual's oral hygiene practices, they can become pathogenic in response to changes in the oral cavity environment [23,24].

As expected, there is a significant variety in the microbiome among different sites within the oral cavity, with notable differences in the gums, tongue, hard palate, soft palate, and tooth surfaces. This makes the oral cavity the site with the highest microbial diversity index after the intestine [25].

During the first two months of a baby's life, bacteria primarily colonize the surfaces of the mucous membranes, and a few months later, with the eruption of teeth, they begin to colonize hard tissues. After this period, the oral microbiome can undergo constant changes throughout all stages of an individual's life [26–28].

Being the main gateway for microorganisms into the human body, both through ingested food and inspired air, oral cavity microorganisms exist in the form of biofilms organized within a complex extracellular matrix. Specifically, microorganisms are involved in organic polymers that are adsorbed to the surface of oral structures, and microbial adhesins allow the attachment of microorganisms to the salivary film, host cells, and exposed dentin. *Porphyromonas gingivalis*, *Streptococcus mitis*, *Streptococcus salivarius*, *Prevotella intermedia*, *Prevotella nigrescens*, *Streptococcus mutans*, and *Actinomyces naeslundii* are some of the oral bacteria possessing these characteristics [29]. This matrix, composed of microbial extracellular products and salivary compounds, helps maintain the ecosystem's balance. Physical, environmental, and biological factors determine the development of these biofilms, which are characterized by their resilience, with some even being resistant to antibiotic penetration and mechanical stress [27,28,30–32].

The Human Oral Microbiome Database (HOMD) reveals that over 80% of the oral microbiome is composed of 200 species from the phyla Firmicutes, Actinobacteria, and Proteobacteria. Along with the genera *Bacteroides* and *Fusobacterium*, these species account for a total of 95% of all identified species in the oral cavity. This study aimed at obtaining a provisional taxonomic scheme; therefore, bacterial isolates came from studies targeting a wide range of individuals and oral health and disease statuses [33]. The genus *Streptococcus* is the most-predominant, followed by *Prevotella*, *Veillonella*, *Neisseria*, and *Haemophilus* [25,34–37].

Some bacteria, particularly *Streptococcus*, produce lactic acid through the fermentation of sugars, which can erode dental enamel. When teeth are not regularly cleaned, dental biofilm accumulates rapidly, potentially leading to tooth decay [25].

Having a crucial role in maintaining oral well-being, if the oral microbiome undergoes minor alterations that create unfavorable conditions, it can become detrimental and lead

to various pathological conditions. Changes in the ecosystem, such as alterations in pH levels and the presence of antibiotics, can trigger these situations. To prevent diseases such as periodontitis or cavities, the microbiome must be in harmony and balance with the host [26–28,38].

In recent times, several theories have been supported by linking periodontitis to other systemic diseases such as cardiovascular diseases. Recent data suggest that the presence of *Porphyromonas gingivalis*, the main causative agent of periodontitis, has been associated with the development of atherosclerosis, regardless of risk factors such as obesity, diabetes, smoking habits, hypercholesterolemia, hypertension, and even a high-fat diet [33,38].

Being a chronic inflammatory disease, periodontitis results from changes in the oral microbiome and causes immune dysregulation and progressive loss of bone mass due to the accumulation of bacteria such as *Porphyromonas gingivalis*. Consequently, there is an increased production of pro-inflammatory cytokines that, upon entering the circulation, can induce an acute response in the liver, thereby increasing levels of C-reactive protein and fibrinogen, which contribute to atherosclerotic events [38,39].

Indeed, studies have already demonstrated that improving oral health leads to a decrease in the progression of cardiovascular diseases. This confirms the role of periodontitis as a significant risk factor to be considered [38,40,41].

### 2.1.2. Nasopharynx

As previously mentioned, each region of the human body provides a different ecological niche that influences the establishment of a normal microbiome, which typically remains constant. In the upper respiratory tract, although the majority of microorganisms inhaled through the air get trapped in the nasal passages and are expelled through nasal secretions, there is a restricted group of microorganisms that colonize the surfaces of deeper mucosal tissues. A prime example of this is the fact that the nasopharynx harbors the highest number of inhaled microorganisms, which become trapped in this region [42].

The microbiome of the nasopharynx contains the same species found in the oral microbiome, along with others such as the genera *Staphylococcus*, *Neisseria*, and *Corynebacterium* [42,43].

In individuals with a considered-normal nasopharyngeal microbiome, it is common to find some potentially pathogenic microorganisms that, however, do not cause host disease. This is because other commensal microorganisms compete with the pathogens, preventing their progression [44]. Additionally, the host's immune system is particularly active on the mucosal surfaces, helping to inhibit the growth of these pathogenic microorganisms. Examples of such microorganisms include *Streptococcus pneumoniae*, *Haemophilus influenzae*, and *Mycoplasma*, which do not have a commensal relationship with the host [45].

In the lower respiratory tract, specifically the bronchi and pulmonary alveoli, it is not common to find bacteria, as they cannot easily reach these areas. If they do manage to reach them, there are first-line defenses, such as alveolar macrophages, ready to avoid any potential colonization [46].

### 2.1.3. Skin

Being the largest organ in the human body, the skin's main function is to serve as a physical barrier, protecting the organism against external aggressions and potential colonization by microorganisms, given that it is in daily contact with a myriad of them [47].

The skin represents a human habitat, where environmental factors and lifestyles can shape the microbial community differently in different specific regions of the human body. One of the main characteristics of the human skin microbiome is its high diversity and interpersonal variation, with a slightly higher concentration in hair follicles. This is extremely important for the health of the skin, as it plays an essential role in its protection against external and potentially harmful substances [15,48,49].

In this way, the low pH of the skin, resulting from the production of acids during keratinization and the excretion of acids by epithelial cells and microorganisms, prevents

new microorganisms from surviving due to those inhospitable conditions. This makes the skin a highly effective barrier against microbial colonization. Additionally, temperature is another factor that can influence the microbiome. Due to the increase in body temperature, sweat production alters the humidity percentage on the skin's surface. The subsequent evaporation of sweat increases the skin's salinity, preventing the proliferation of certain microorganisms, especially Gram-negative bacteria [47].

In this way, by combining all the described factors with the various microenvironments, we have different microbial populations in the various regions of the skin.

The cutaneous microbial community is specific to each region of the human body, each individual, remaining quite stable over time, despite constant exposure to different environments. Instead of acquiring the prevalent microorganisms from the environment, hosts maintain their microbiome, which ends up functioning as a unique microbial fingerprint for each person [50,51]. Grice and Segre published that the four dominant phyla residing on the skin are Actinobacteria, Proteobacteria, Firmicutes, and Bacteroidetes, the most-predominant bacterial genera being *Staphylococcus*, *Cutibacterium*, and *Corynebacterium*, each of which is concentrated differently among various skin locations [48].

Areas with a high density of sebaceous glands, such as the back, face, chest, and the area behind the ears, tend to be colonized by large quantities of lipophilic microorganisms, such as *Cutibacterium*. This is because these bacteria hydrolyze the triglycerides found in the sebaceous glands, subsequently releasing free fatty acids on the skin, which acidify and soften it, acting as an emollient. Moreover, these areas tend to have lower levels of diversity compared to dry and moist regions [48,52,53].

The role of sebum in defining the cutaneous microbiome can be reflected in age-related changes in the composition and diversity of this microbiome. After birth, babies are colonized by more bacteria of the phylum Firmicutes than Actinobacteria. Generally, as sebaceous glands mature during puberty, there is a shift in the skin microbiome with an increase in Actinobacteria, including *Corynebacterium* and *Cutibacterium*. This age-associated change may be a significant factor in reducing the incidence and severity of common childhood skin diseases, such as atopic dermatitis. The fungi present in sebaceous areas tend to be less diverse than the bacterial communities, with the dominant species being *Malassezia* [54,55].

Humid or occlusive areas of the skin typically have a microbial community composed of Gram-positive bacteria such as *Staphylococcus* and *Corynebacterium*, which prefer high moisture concentrations in their habitat, as found in the belly button, armpits, groin, behind the knee, and inner elbow areas. In these locations, the high humidity, elevated temperature, and concentration of skin lipids promote the growth of the microbiome. Bacteria of the *Staphylococcus* genus occupy an aerobic niche, and it is believed that they utilize the urea present in sweat as a source of nitrogen [56].

The areas that show the greatest diversity are dry regions such as the forearm, elbows, knees, buttocks, and various parts of the hand. In these areas, there is a predominance of large quantities of Proteobacteria and Bacteroidetes, but also some Firmicutes and Actinobacteria, in comparison to the humid and sebaceous zones. It is in these areas that some skin disorders such as psoriasis are observed, which result from a failure in immune tolerance to the microbiome [53,57].

The most-predominant microbial species on the skin are Gram-positive bacteria, with *Staphylococcus epidermidis* being particularly prominent, constituting almost 100% of the bacterial community on this surface. This bacterium is mostly harmless; however, when invading other regions, it can become harmful, namely in the case of catheters; when they are introduced into the body, *S. epidermidis* can invade the host and cause nosocomial infections by forming a biofilm that protects it from immune responses and antibiotic action [48].

The use of antibiotics, cosmetics, and personal hygiene products, as well as lifestyle and dietary habits are factors that can also influence the skin microbiome. While most of these microorganisms are harmless or beneficial, some have recently been associated with

skin conditions such as acne, psoriasis, and eczema [51]. Thus, the study of the variability of these microbial communities in different areas of the human body has helped to understand why eczema tends to manifest in humid areas, such as the folds of the arms and legs, and why psoriasis occurs in drier areas, such as the elbows and knees [53,58].

Among epithelial surfaces, the skin is unique in its complex ecological interactions with the environment [59,60]. Several external and host factors affect the microbial composition, namely temperature, humidity, and light exposure. Gender, genotype, immune status, and the use of cosmetics can also affect the microbiota. The microbial composition such as the population size and community structure is affected, as represented in Figure 1 [48,61]. The immune system of the host constantly adjusts itself, actively participating in the formation of a consistent and location-specific skin microbiome as individuals reach adulthood [62]. Moreover, the competition that occurs both within and among different microbial species plays a crucial role in the development and sustainability of a well-functioning microbiome [59]. Puberty has a profound effect on the composition of the skin microbiome. Sex hormones trigger the increased activity of the sebaceous gland and apocrine gland, resulting in increased skin lipids, which favor the colonization and growth of lipophilic bacteria, including *Cutibacterium acnes* [62].

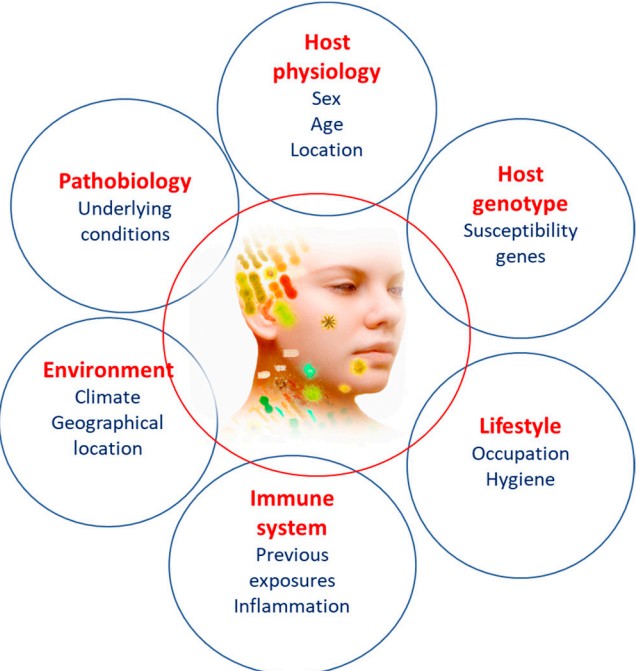

**Figure 1.** Factors contributing to skin microbiome variation.

### 2.1.4. Gastrointestinal Tract

The large and diverse colonization of the human gastrointestinal tract begins early in life. The gastric and intestinal bacterial ecosystem of an adult contains a complex array of microorganisms with over 100 trillion microbial cells, including more than 1000 bacterial species [63].

The colonization of the gastrointestinal tract begins shortly after birth, through maternal contact. Over the first few weeks, it becomes more diverse and changes until it stabilizes around 2 to 3 years of age, by which time it is already very similar to the gastrointestinal microbiome of an adult [64].

Regarding the stomach's microbiome, the fact that this organ has a highly acidic pH and secretes pepsinogen in its mucosa prevents any potential colonization, as microorganisms are destroyed. As a result of these adverse environmental conditions, the population remains very low. Some species resistant to the stomach's acid, hydrochloric acid, are

included in the *Lactobacillus* and *Streptococcus* genera; *Heliobacter pylori* is also resistant. However, the latter can cause chronic active gastritis, peptic ulcers, or even neoplasms [57].

The intestinal microbiome has several functions and interacts with the host beyond being merely a physiological support in the food digestion process. It is part of and regulates the intestinal mucosal barrier, controls nutrient absorption and metabolism, aids in the maturation of immune tissues, and prevents the proliferation of microorganisms. Under physiological conditions, the microbiome continues to stimulate the immune system since it is an effective defense mechanism against foreign agents in the body [65–67].

The composition of the intestinal microbiome is extraordinarily diverse and dynamic in short periods because the intake of food or medications significantly affects the microbial community. However, its composition appears to remain stable over time among individuals and their closest family members. Despite the daily intake of microorganisms through food, this population remains relatively constant, and exogenous factors are needed to disturb the balance of the commensal microbiome, such as the presence of pathogenic bacteria or the use of antibiotics [68–70].

The majority of nutrients that we obtain from the food we eat go through various human enzyme processing before they get absorbed in the small intestine. However, the intestinal microbiome contributes to the metabolism of fibers that are typically not digested by these enzymes. In the large intestine, a group of microorganisms, including *Lactobacillus* and *Bifidobacterium*, contribute to the metabolization of plant-derived polysaccharides, fibers, oligosaccharides, undigestible proteins, and intestinal mucins to provide energy sources for the host [71–74].

In addition to its role in digesting certain foods, the intestinal microbiome plays a vital role in preventing the invasion of certain pathogens by creating microbial resistance. It contributes to the education and stimulation of the immune system, maintains the integrity of the intestinal epithelium and homeostasis, and enhances the motility of the gastrointestinal tract. The bacteria that make up the colon's microbiome are also responsible for synthesizing vitamins such as B12, K, biotin, thiamine, and folic acid [68,75].

The majority of microorganisms that make up this community are anaerobic bacteria from the phyla Bacteroidetes and Firmicutes [76], which play a fundamental role in food digestion and occupy about 90% of the gastrointestinal microbiome. Additionally, there are microorganisms from the phylum Actinobacteria (family Bifidobacteriaceae) and Proteobacteria (family Enterobacteriaceae), although in much smaller quantities. Other bacteria from different phyla can also be found, such as Fusobacteria and Euryarchaeota (Kingdom Archaea), representing a small percentage of the intestinal microbiome. Intestinal peristalsis, stomach-derived hydrochloric acid, and the high concentration of bile salts result in a low count of microorganisms in the small intestine. As one progresses from the duodenum to the ileum, bacterial density increases. At the beginning of the small intestine, the microbiota is very similar to that present in the stomach, while in the terminal part of the small intestine, the microbiota closely resembles that of the large intestine [77].

The bacteria most frequently found in the duodenum and jejunum belong to the genera *Streptococcus* and *Lactobacillus*, which is very similar to the microbiome found in the stomach. This similarity is because these structures are located nearby, and in the duodenum, there are bile and pancreatic secretions responsible for the acidity [78]. The following portion, the ileum, already presents a moderate microbiome with a predominance of the genera *Enterococcus*, *Lactobacillus*, *Bacteroides*, and *Bifidobacterium*. In the colon, a larger and more-complex population of anaerobic microorganisms can be found, such as *Bacteroides*, *Bifidobacterium*, *Eubacterium*, and *Clostridium*. This is due to a progressive decrease in acidity, allowing for the existence of more microorganisms. Consequently, the colon exhibits greater microbial diversity among individuals, making it an ideal site for fermentation between the microbiome's microorganisms and nutrients from food digestion [75,78]. In Table 1, the composition and concentrations of more-important microbial species in different locations of the gastrointestinal tract are depicted based on several authors [79–82].

**Table 1.** Composition and luminal concentrations of important microbial species in gastrointestinal tract regions [79–82].

| Region | Luminal Concentration | Main Composition |
| --- | --- | --- |
| Stomach | $0–10^2$ | *Lactobacillus*<br>*Candida*<br>*Streptococcus*<br>*Helicobacter pylori*<br>*Peptostreptococcus* |
| Duodenum | $10^2$ | *Lactobacillus*<br>*Streptococcus* |
| Jejuno | $10^2$ | *Lactobacillus*<br>*Streptococcus* |
| Proximal ileum | $10^2$ | *Lactobacillus*<br>*Streptococcus* |
| Distal ileum | $10^7–10^8$ | *Clostridium*<br>*Steptococcus*<br>*Bacteroides*<br>*Actinomycinae*<br>*Corynebacteria* |
| Colon | $10^{11}–10^{12}$ | *Bacteroides*<br>*Clostridium*<br>*Bifidobacterium*<br>*Enterobacteriaceae* |

All these bacteria are usually commensal, but in immunocompromised or weakened individuals, they can become pathogenic, even in small quantities. It has been concluded that the origin of a disease is not a specific microorganism, but rather an imbalance in the intestinal microbiome, which may involve multiple microorganisms at the same time, referred to as dysbiosis [83]. This has implications for human health, leading to a variety of diseases such as inflammatory bowel diseases, cancers, metabolic disorders, and even cardiovascular diseases, which is the main subject of this review [84].

### 2.1.5. Urogenital Tract

The urinary tract and the genital tract are intrinsically linked, so they are often referred to as the urogenital tract. The urinary tract is usually considered a sterile space and exhibits resistance to bacterial colonization, even though the distal urethra often encounters frequent contamination. However, recent technological advancements have allowed us to gain new insights into this topic. It is now known that there is also a resident microbiome in the bladder, which, when in balance, has no negative impact on the body [85,86].

The main defense against urinary tract infections is the flow of urine, especially during complete bladder emptying, where bacteria invading the bladder are eliminated through urination. Additionally, urine possesses other antimicrobial defense mechanisms such as a high concentration of urea, acidic pH, and various immune barriers. Besides this defense, the protection of the upper organs of the urinary system against colonization by bacteria from the vagina or penile surface is ensured by the action of a sphincter in the urethral area, which acts as a barrier to the entry of microorganisms [85,87].

It is estimated that the diversity of the urinary microbiome is very comparable to the diversity of the vaginal microbiome, being mostly composed of *Lactobacillus*, but also some *Staphylococcus*, *Streptococcus*, and possibly, a residual amount of *E. coli*, mainly present in the distal urethra. However, there may also be transient colonization by fecal microorganisms such as enterobacteria, which can lead to infections of the upper urinary tract [88,89].

Just like in the intestinal microbiome, the urinary microbiome plays an important role in maintaining the balance of the urinary system. Some of the most-important functions include the degradation of toxic compounds, the maintenance of mucosal integrity, the activation of the immune system, and the production of antimicrobial compounds [87,90].

As concerns the genital tract, the vaginal microbiome has long been considered an important defense mechanism against infections in women. Subject to various modifications due to age and hormonal factors in women, the vagina exhibits a great microbial diversity. Before puberty, this microbiome consists mostly of *Streptococcus*, *Staphylococcus*, and *Escherichia coli*. Upon puberty and with the production of estrogen, *Lactobacillus* becomes the predominant species, and the continuous production of lactic acid is responsible for vaginal acidification, preventing the colonization of new microorganisms, whether bacteria, fungi, or viruses. Simultaneously, these microorganisms will produce antimicrobial substances such as hydrogen peroxide, which inhibits the spread of viruses and the growth of bacteria and fungi. They will also stimulate the vaginal immune system, enhancing local defense mechanisms, and form a biofilm that prevents pathogenic agents from accessing the vaginal mucosa. During this phase, it is common for the microbiome to include *Lactobacillus*, *Corynebacterium*, *Staphylococcus*, *Streptococcus*, and *Bacteroides* [1].

During pregnancy, the microbiome is characterized by an increase in *Lactobacillus* species, remaining relatively stable and less subject to variations, also due to the production of lactic acid, considered a defense mechanism. In menopause, there is a decrease in the production of estrogen and glycogen, causing the vaginal microbiome to have a pH of around 7, and the composition becomes very similar to that seen in the pre-pubertal period, predominantly colonized by *S. epidermidis*, *Streptococcus*, and *E. coli* [91,92].

In a general perspective, both disorders in the genital and urinary microbiome share the same risk factors, such as the use of antibiotics, unprotected sexual intercourse, the use of diaphragms, and sexual promiscuity, among others, which can alter the microbiota and lead to possible infection. Infections are more likely to occur in women than in men because women lack certain protective factors, such as the distance between the two anatomical structures, the length of the male urethra, and the antimicrobial activity of prostatic fluid.

## 3. Microbiome Evolution throughout Life

During pregnancy and up to the moment of birth, the human being is composed only of its somatic cells and is kept in a sterile environment. However, after birth, the newborn is exposed to a wide variety of microorganisms such as bacteria, fungi, and viruses, among others, many of which are provided by the mother during and after passage through the vaginal canal, an ecosystem heavily colonized by a relatively limited set of bacteria, or shortly after a cesarean section by microorganisms from the mother's skin. Thus, early on, the construction of a positive microbiome for future life begins [1,19,45,68].

A new microbial ecosystem begins to form in the body of the newborn, developing from the first days of life and becoming essential in the processes of immune recognition and tolerance. Its profile depends on the type of delivery (vaginal or cesarean) and the feeding method (breastfeeding or formula feeding), creating a wide range of microbiome compositions. Therefore, the interindividual variation in intestinal microbial diversity is greater among infants than among adults [64,68].

The beginning of human microbiome development starts during birth, with the colonization of microorganisms from the environment. However, there are recent studies defending that colonization starts before birth [93,94]. In the first hours of life, the microorganisms present in the mother's vaginal and fecal microbiota are typically the most important sources of inoculation. The type of delivery can lead to differences in microbiome development, which may later contribute to variations in the host's normal physiology or even predisposition to diseases [68].

When a normal delivery occurs, the newborn will be colonized by bacteria present in the mother's vaginal canal, such as *Lactobacillus*, *Prevotella*, and *Sneathia*. On the other hand, in the case of a cesarean section, the microorganisms that will colonize the newborn will be similar to those found on the mother's skin, namely *Staphylococcus*, *Corynebacterium*, and *Cutibacterium*. As a result, bacterial communities only differentiate over the years with the emergence of new species and their evolution, making them distinct from one another [95,96].

During the first months of a lactose-based diet, different formula feedings will influence the quantity and diversity of microorganisms found in the newborn's microbiome. Breastfeeding predominantly fosters bacteria such as *Staphylococcus*, *Streptococcus*, *Clostridium*, *Bacteroides*, and *Bifidobacterium*. The latter is responsible for protecting the baby, as they are highly adapted to process the milk's oligosaccharides, metabolizing sugars into acids and contributing to competitive inhibition with other bacteria by adhering to the intestinal mucosa. This protects against intestinal pathogens [97]. On the other hand, the number of *Bifidobacterium* is significantly lower in infants fed with formula, whereas other bacteria such as *Escherichia coli* and *Bacteroides* predominate. Thus, it can be said that breast milk is fundamental in preventing dysbiosis and potentially more-complicated diseases in the future [77].

Not only do these two factors influence the type of microbiome found in newborns, but depending on the geographical region where a certain baby is located, they will present a completely different microbiome, reflecting the environmental impact and sanitary conditions of the area. The use of antibiotics also heavily influences the infant's microbiome, creating significant disparities among infants [75,98].

The introduction of solid foods and weaning signify a shift in the ecosystem of microorganisms, leading to an increasing resemblance to the microbiome of an adult. Thus, in the first year of life, the gastrointestinal tract progresses from a nearly sterile state to an extremely dense colonization, ending with a community very similar to that found in the adult gastrointestinal tract. The microorganisms to which the child will be exposed are, therefore, essential for the maturation of his/her microbiota [99].

During puberty, changes in sebum production occur simultaneously with an increase in the levels of lipophilic bacteria on the skin surface. Physiological and anatomical differences between individuals of the female and male sexes, such as sweat, sebum, and hormone production, are contributing factors to the significant changes observed between genders during puberty [100]. As age advances, significant changes occur throughout the body, and the microbiome is no exception. The number of microorganisms present in the body begins to decrease drastically, especially anaerobic bacteria in the intestine, such as *Bifidobacterium*, which act as protectors of the intestinal tract. Additionally, there is a decrease in the production of short-chain fatty acids and an increase in proteolytic activity, compromising the functioning of the digestive system. Moreover, with aging, there is also a significant increase in enterobacteria, which can be considered pathogenic [77].

## 4. Factors Influencing Microbiome

The colonization process begins early and stabilizes around the age of 3 years, but several factors can influence the microbiome, leading to beneficial or detrimental changes. Therapeutic interventions, diet, genetics, antibiotic use, hygiene conditions, lifestyle, and dietary habits are some of these factors that are crucial for the populations of bacteria present in the body [76]. The plasticity of the microbiome has been implicated in numerous pathological conditions, and an unfavorable alteration of the resident microbiome is referred to as dysbiosis, which can be caused by various factors such as diet, increased stress or inflammatory markers, and the use of antibiotics. This partial or complete alteration of the human microbiome can disrupt the ecosystem's homeostasis, leading to a new state characterized by the emergence of various pathological factors in the host, such as potentially harmful bacteria. These changes may, thus, explain why some individuals are more prone to developing certain diseases [2,77].

A healthy lifestyle is essential for preserving the microbiome and human health. Therefore, sedentary behavior, stress, smoking, and dietary excesses also jeopardize the body's balance. Recent lifestyles, particularly changes in the Western diet, which now includes higher levels of fat and sugar, can influence the structure and activity of resident microorganisms. Diet-induced alterations are suspected to contribute to the development of chronic diseases, including cardiovascular diseases and obesity [78]. According to David

et al., even after a short period of consuming a different diet than usual, changes in the composition and activity of the intestinal microbiome can be observed [79].

The intake of nutrients such as proteins, lipids, carbohydrates, polyphenols, prebiotics, and probiotics can alter the intestinal microbiome, leading to biological effects, including changes in metabolism, the immune system, and the production of pro- and anti-inflammatory metabolites. These modifications can result in human diseases such as cardiovascular diseases, type 2 diabetes *mellitus*, obesity, metabolic syndrome, and skin and autoimmune diseases [80].

Extrinsic factors influencing the microbiome will be discussed briefly.

### 4.1. The Effects of Diet

Eating habits play a crucial role in shaping the composition and diversity of the gut microbiota, ultimately impacting our overall well-being. Remarkable discoveries provide significant health advantages, but also underscore the therapeutic potential inherent in the food industry. This represents a significant milestone in healthcare, as a mere shift in our dietary choices can potentially reduce the financial burden of medical treatments. Contemporary diets, particularly those prevalent in Western societies, have been linked to the development of numerous preventable conditions such as asthma, obesity, and multiple sclerosis [101]. Optimal microbiota is attained when microbial members possess the capability to metabolize sugars and when the microbiota undergoes an adaptation in response to the available nutrients in the intestinal environment [102]. The beneficial effects of fiber are well documented in diverse studies [103,104]. The consumption of a diet high in fat has been shown to have adverse consequences on the permeability of the mucus layer and can also hinder metabolic processes [105]. The Mediterranean diet is considered beneficial to health [106]. According to research on micronutrients, gluten-free bread can help alleviate microbiota dysbiosis, but further investigation is needed to gain a deeper understanding of these hypotheses [107].

### 4.2. Antibiotics and Drugs

The effects of antibiotics can be attributed to various factors, including their mechanism of action, antibiotic class, the level of antibiotic resistance, and the dosage administered during treatment. Furthermore, elements such as how a substance is delivered, its pharmacokinetics and pharmacodynamics, and the extent of its activity (whether it affects a wide or limited range of microbes) all influence the gut microbiota [108]. Non-steroidal anti-inflammatory drugs are commonly used in daily life; however, they can be causative agents for stomach ulcers, with metabolic disorders being extensively researched in this context [109]. Several other medications, including proton pump inhibitors (PPIs), antidepressants, metformin, laxatives, and oral steroids, have been studied for their significant impacts on gut dysbiosis [110].

### 4.3. Oxidative Stress

A link between increased oxidative stress and a decrease in the diversity of gut microbiota has been established [111]. The modern dietary pattern, often referred to as the Western-style diet, is characterized by its high content of fats and refined sugars. Consuming these substances in large quantities leads to increased inflammation and the production of reactive oxygen species (ROSs). Subsequent ROS production triggers the inflammatory cascade [112]. The stress induced by the generation of ROSs is defined as oxidative stress, which encompasses biological processes for detoxification and repairing secondary damage [113].

### 4.4. Socioeconomic Status

The gut microbiota composition in infants is primarily influenced by the economic status of the region or country in which they are born and raised. This is because the food choices available to mothers significantly impact how they can nourish their infants. It

is scientifically accepted that people suffering from malnutrition, including undernourished infants, are more susceptible to various health issues. Undernourished infants, in particular, are prone to developing a condition known as dysbiosis, characterized by an overabundance of enteropathogens such as Enterobacteriaceae [114].

## 5. Microbiome Influence on Atopic Dermatitis

The influence of the microbiome on atopic dermatitis (AD) will be detailed in the following section.

### 5.1. Atopic Dermatitis

AD, also known as atopic eczema, is a chronic inflammatory skin disease characterized by dry and red skin, intense itching, and recurrent scaly lesions with a typical distribution [58,115–118]. The pathogenesis of the disease is not fully understood; however, it appears to result from a complex interaction between defects in the skin barrier function, environmental and infectious agents, and immune system dysregulation [119]. Although it mostly begins in childhood, it is also prevalent in adults, being the leading non-fatal health problem attributed to skin diseases, inflicting a significant psychosocial weight on patients and their families, elevating the likelihood of conditions such as food allergies, asthma, allergic rhinitis, and other immune-related inflammatory ailments, along with mental health issues [120]. Therefore, it carries a relevant weight in healthcare systems [115]. AD is classified as mixed or pure. Mixed AD represents approximately 60% of cases and is associated with respiratory symptoms. On the other hand, pure AD is classified when there is no associated respiratory disease [121].

The symptoms of AD differ depending on the age group. Infants commonly experience AD symptoms on the scalp, face, neck, trunk, and the outer parts of the limbs. In children, the areas affected usually include the inner parts of the limbs such as the elbows, knees, neck, wrists, and ankles. During adolescence and adulthood, the inner parts of the limbs, hands, and feet are frequently more affected. Regardless of age, the itchiness associated with AD typically persists throughout the day and intensifies at night, causing sleep deprivation and significantly reducing the quality of life [120].

In all stages of the disease, the lesions can present as acute, subacute, or chronic eczematous. These stages can occur sequentially, although in most cases, they coexist in different regions of the body or even in the same region, indicating the characteristic episodic evolution of the condition [121].

Due to similarities in some symptoms, it can sometimes be challenging to differentiate AD from other skin conditions (such as seborrheic dermatitis, contact dermatitis, psoriasis, or scabies). Nevertheless, in many instances, the diagnosis is aided by the presence of atopy in the family history and the pattern of lesion distribution [119].

### 5.2. Dysbiosis in AD Patients

The exact cause of AD is not completely understood; dysbiosis of the skin microbiota is believed to play a significant role in its development and exacerbation. Therefore, the molecular mechanisms of the underlying dysbiosis of the skin microbiota in atopic dermatitis will be briefly described [122]. Dysbiosis in AD is often associated with a decrease in microbial diversity on the skin, usually referred to as altered microbial diversity. The reduced diversity is thought to be driven by immune dysregulation in AD patients [123]. An overactive immune response can create an inhospitable environment for beneficial bacteria. AD is characterized by an abnormal immune response, including increased levels of pro-inflammatory cytokines such as interleukin-4 (IL-4), interleukin-13 (IL-13), and interleukin-22 (IL-22) [124]. These cytokines contribute to skin inflammation and barrier dysfunction. In response to inflammation, the skin may produce antimicrobial peptides (AMPs) such as defensins and cathelicidins, which can disrupt the balance of skin bacteria [125]. These peptides are part of the skin's innate immune defense system, but can also harm beneficial bacteria. An altered lipid composition is also important to

AD development. The skin's lipid barrier, composed of lipids such as ceramides, plays a critical role in maintaining the skin's integrity and preventing water loss. In AD, there is a disruption in the composition of these lipids that affects the growth and survival of certain skin microorganisms that metabolize specific lipid substrates, leading to imbalances in the skin microbiota [126]. *Staphylococcus aureus* is often found in higher abundance on the skin of AD patients compared to healthy individuals. This bacterium can produce toxins and virulence factors that exacerbate skin inflammation. *Staphylococcus aureus* can outcompete other commensal bacteria in AD patients, further reducing microbial diversity [15]. A compromised skin barrier, also called an impaired skin barrier, is a hallmark of AD. It allows for increased transepidermal water loss and the penetration of allergens, irritants, and microbes into the skin. The damaged skin barrier can create an environment that is conducive to the growth and colonization of pathogenic bacteria while hindering the growth of beneficial microbes [15]. Genetic factors also play a role in AD susceptibility. Mutations in genes related to skin barrier function and immune regulation, such as filaggrin (FLG) and various cytokine genes, can increase the risk of AD and contribute to dysbiosis [127].

In summary, dysbiosis of the skin microbiota in atopic dermatitis is a complex interplay of altered microbial composition, immune dysregulation, changes in skin lipids, and impaired skin barrier function. These molecular mechanisms collectively contribute to the chronic inflammation and skin manifestations seen in AD. Understanding these mechanisms is crucial for developing targeted treatments to restore the balance of the skin microbiota and alleviate symptoms in AD patients.

The mechanisms of the skin microbiota's influence on AD pathogenesis are depicted in Figure 2.

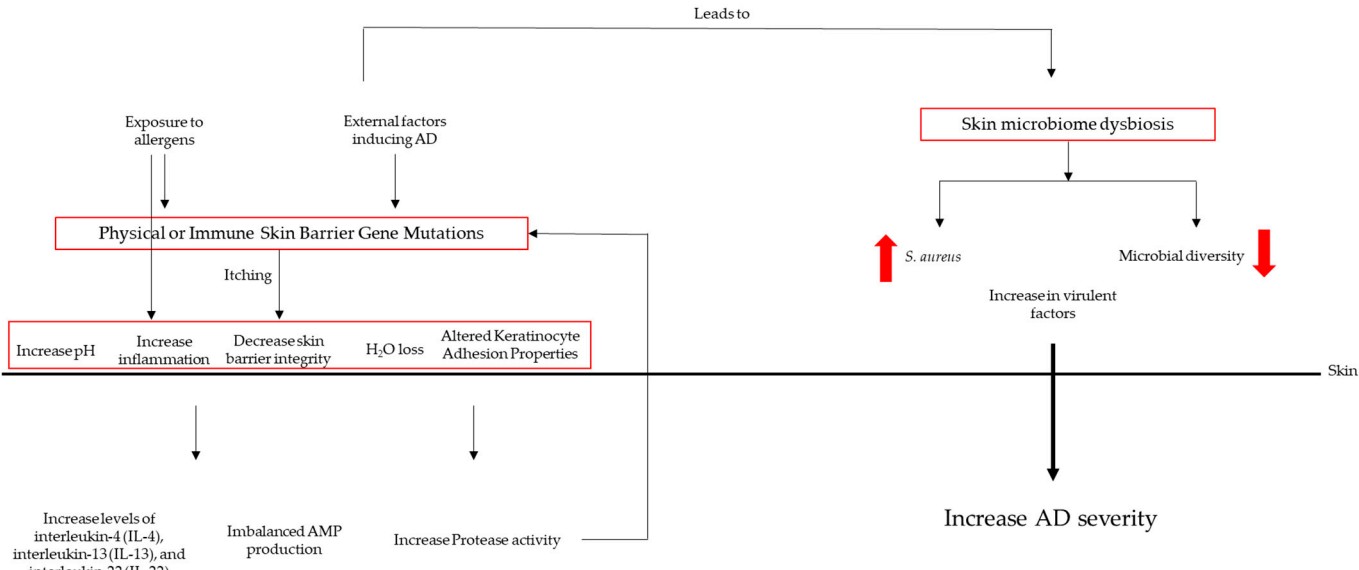

**Figure 2.** Mechanisms of skin microbiota's influence on AD pathogenesis.

### 5.3. Microbial Colonization

Profound changes in the skin microbiota occur in some patients with AD, and the pathogenic importance of microbial organisms is recognized [128]. The diversity of the microbiome decreases in inflamed atopic skin, with a reduction in the genera *Streptococcus*, *Corynebacterium*, and *Cutibacterium*, as well as the phylum Proteobacteria, in favor of an increase in the genus *Staphylococcus*, particularly *S. aureus* [129].

In AD, there is a notable loss of strictly anaerobic bacteria, altering the skin's microbiome from anaerobic to aerobic metabolism. Healthy skin is normally devoid of oxygen, but dry and scaly skin with compromised epidermal barrier function can increase oxygenation and decrease the abundance of strictly anaerobic bacteria such as *Lactobacillus*

spp. or *Finegoldia* spp. In the absence of oxygen, skin bacteria metabolize organic substances through fermentation, such as the amino acid serine derived from the breakdown of filagrine, producing lactic acid, propionic acid, and other short-chain fatty acids. These metabolic products reduce the skin's pH to below 5.5, preserving the skin's protective acidic environment. Additionally, Gram-positive anaerobic cocci such as *Finegoldia*, *Anaerococcus*, and *Peptoniphilus* stimulate a rapid antimicrobial peptide (AMP) response in human keratinocytes, which could be an important signaling mechanism for keratinocytes when the skin is damaged. When these organisms are partially or completely absent, the signaling mechanisms in keratinocytes and other protective functions may be compromised, increasing the likelihood of *S. aureus* colonization [130].

Colonization by *S. aureus* on AD skin has been directly linked to disease severity, but, as mentioned earlier, the role of other constituents within the skin's bacterial community may be just as significant [115]. Dysbiosis related to AD is often characterized by colonization by *S. aureus* and the simultaneous loss of other potentially beneficial species, although the loss of anaerobes in AD does not seem to be driven by the presence of *S. aureus* [130].

The skin of AD harbors a microbial growth environment that is very different from that of normal skin, and this may be crucial in explaining the dysbiosis observed in AD. A dysfunctional physical barrier of the skin leads to an increased pH on the skin surface, which favors the growth of *S. aureus* [115]. Besides the frequent presence of *S. aureus* on the skin of people with AD, there are other factors that strengthen the idea that the microbiota has a significant impact on the development of the disease [58].

### 5.4. The Role of S. aureus in Atopic Dermatitis

AD has a known relation with altered skin microbiota, with a high prevalence of colonization by *S. aureus* and secondary infections that were recognized long before the application of DNA sequencing approaches [129]. *S. aureus* colonizes approximately 9 out of 10 patients with AD [119], and it can initiate or exacerbate skin diseases either through barrier defects or altered immunity [58].

*S. aureus* is detectable in more than 90% of AD skin, although it is less frequently detected in healthy skin [131]. AD has been associated with a notable rise in the prevalence of *S. aureus* and a decline in anaerobic species [130], and it can act as a persistent allergen, stimulating the production of IgE antibodies, or as an irritant with inflammatory potential when colonizing atopic skin [128]. Beside the awareness of *S. aureus*'s capacity to induce inflammation and the occurrence of dysbiosis in different skin conditions, it remains uncertain whether these alterations are a result of the disease itself or if *S. aureus* plays a role in initiating the disease [58].

In AD, *S. aureus* dominates the microbial landscape and negatively correlates with various skin commensals such as *Staphylococcus epidermidis* and *Corynebacterium* spp., thus potentially eliminating the regulatory or protective potential of these microorganisms [130]. The quantity of *Staphylococcus* spp., particularly *S. aureus* and *S. epidermidis*, is higher during the eruption period (episodic exacerbation) of AD compared to the post-eruption period. However, individuals with more-severe exacerbations are colonized with dominant strains of *S. aureus*. The correlation of *S. aureus* with AD results in the exacerbation of the pathology [58]. It is important to note that *S. epidermidis* may limit the growth of *S. aureus*, and the severity of the disease is inversely correlated with the abundance of *S. epidermidis* versus the abundance of *S. aureus* [130].

There is a sophisticated relationship between the host and *S. aureus*, where host factors, including the hostile environment created by the physical, chemical, and antimicrobial properties of healthy skin, may be altered in AD skin, facilitating colonization. Specific pathogen factors include highly evolved mechanisms that facilitate adhesion, epidermal invasion, and pro-inflammatory mechanisms, promoting or exacerbating the inflammatory component of AD [129].

The prevalence of *S. aureus* carriers in patients with AD is about 70% on lesional skin compared to about 39% on non-lesional or healthy skin in the same patient [129]. There is

a higher abundance of *Staphylococcus* spp. at 2 months in newborns who do not develop AD by 1 year of age compared to those who do develop AD by 1 year. This suggests that exposure to *Staphylococcus* spp. at an early age is beneficial for proper immune system education [58].

*S. aureus* plays a highly influential role in the pathogenesis of AD, is associated with severe disease flares, and significantly influences the disease phenotype [129].

*5.5. Gut Microbiota in AD Patient*

In a typical healthy gut, Firmicutes, Bacteroidetes, Actinobacteria, and Proteobacteria are present. These categories are quite consistent overall, but the specific types of bacteria within them can vary from person to person.

Xue et al. discovered that certain groups of bacteria, such as Tenericutes, Mollicutes, Clostridia, Bifidobacteriaceae, Bifidobacteriales, Bifidobacterium, Christensenellaceae R7, Bacilli, and Anaerostipes, were linked to a reduced risk of developing atopic dermatitis (AD) [13]. On the other hand, bacteria groups such as the Eubacterium hallii group, Clostridiaceae_1, Bacteroidaceae, Bacteroides, Anaerotruncus, an unknown genus, and Lachnospiraceae UCG 001 appear to be potential factors that could increase the risk of AD [13].

Park et al. noticed differences in the gut bacteria of children with atopic dermatitis (AD). Children with transient AD had lower levels of *Streptococcus* bacteria, but higher levels of *Akkermansia*. Conversely, children with persistent AD had the opposite pattern [132]. Moreover, in children with AD, the Clostridium genus became more abundant. There were also changes in the functional genes of their gut bacteria related to energy metabolism and short-chain fatty acid (SCFA) production, with a decrease in these functions. This conclusion was supported by another study by Lee, who used metagenomic analysis and arrived at a similar finding [133].

Yap et al. showed that, in children with early-stage AD, higher levels of *Enterococcus* and *Shigella* were present. Interestingly, these higher levels of these bacteria have been linked to increased production of serotonin, which, in turn, worsens skin pigmentation. Furthermore, the abundance of *Bifidobacterium*, a beneficial type of bacteria, tends to decrease in these children [134].

Several other studies are being conducted to clarify the relation between the gut microbiota and AD, enhancing the knowledge about the microbiota profiles in AD patients.

Several studies are confirming the idea that the immune responses in AD are influenced by the gut microbiota [135]. Figure 3 summarizes this information and focuses on three key aspects: (1) the anti-inflammatory effects of short-chain fatty acids (SCFAs), (2) the impact of tryptophan metabolites on the aryl hydrocarbon receptor (AHR) pathway, and (3) the role of toll-like receptor signaling, which can be influenced by the gut microbiota and is highly relevant to AD. Briefly, the gut microbiota, in particular Bifidobacteria, produces short-chain fatty acids, which activate SCFA-sensing G-protein-coupled receptors and inhibit histone deacetylases. These phenomena reduce inflammatory responses and promote TH1/TH2 equilibrium. This equilibrium is also helped with the production of D-tryptophan, which, in turn, is a microbial metabolite, which can also activate aryl hydrocarbon receptors, inhibiting inflammatory responses and promoting the skin barrier. TH1/TH2 balance is also promoted by molecular patterns associated with the presence of pathogens produced by the gut microbiota.

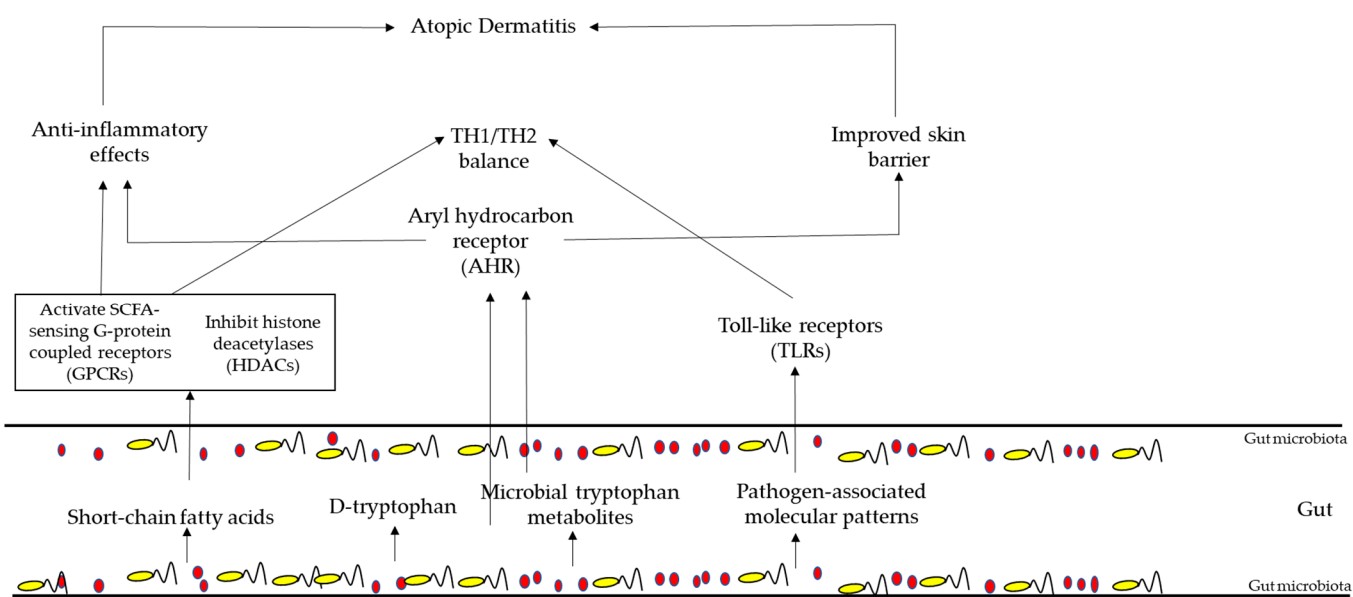

**Figure 3.** AD pathogenesis regulation by the gut microbiota.

## 6. Therapies Targeting Gut Microbiota's Composition

The manipulation of the host's microbiome has shown numerous promising applications in various fields of science and medicine. In addition to the therapeutic strategies already in use, modulating the human microbiome through the use of probiotics and prebiotics, adopting a Mediterranean diet, and fecal microbiota transplantation are being considered as alternatives to antibiotic therapy to enhance beneficial functions in dysbiosis treatments in several situations [52]. In Table 2, therapies targeting the microbiota's composition to promote a healthy status are exemplified.

**Table 2.** Therapies targeting microbiota's composition.

| Therapy | Definition | Examples | Reference |
|---------|------------|----------|-----------|
| Probiotics | Beneficial living microorganisms capable of establishing themselves in the human gut to foster or reinstate a well-balanced gut microbiota composition | *Lactobaccillus* strains *L. reuteri* (microencapsulated in yogurt) *L. plantarum* (capsules) | [136–138] |
| Prebiotics | Nutritional elements that nourish and encourage the development of a thriving gut microbiota composition | Plant polyphenols Fruits and vegetables (e.g., apples) Dietary fructans Foods high in inulin and/or oligofructose | [138–144] |
| Diet intervention | Ingestion of dietary components serving as sources of energy and nutrients for microbial growth, shaping the microbial community composition | Diets rich in fiber and vegetables | [145–147] |
| Fecal microbiota transplantation | Transference of fecal matter, containing a mixture of beneficial microorganisms, from a healthy donor to the gastrointestinal tract of a recipient | Restore the microbial balance in the intestine | [148–150] |

### Therapies Targeting Microbiota Composition in Atopic Dermatitis

The factors that lead to the onset of allergic diseases during infancy are not completely comprehended at present. One well-accepted theory centers on the gut microbiota, wherein the mix and characteristics of beneficial bacteria interact with the evolving immune system. These interactions have the potential to impact immune development, potentially resulting

in Th2 allergic responses. Consequently, there has been substantial scientific inquiry into preventative or therapeutic approaches targeting the gut microbiota [151,152]. Intestinal permeability is increased in patients with AD. In recent years, additional data have emerged regarding the benefits of treatment with next-generation microbiome-based biotherapies in patients with AD [152].

Traditionally, treating *S. aureus* infection in AD involved using broad-spectrum antibiotics to decolonize, but this approach carries the risk of promoting antibiotic resistance and disturbing the balance of beneficial microbes. Additionally, *S. aureus* has developed mechanisms of antibiotic resistance. One potential method that was explored for managing *S. aureus* abundance in AD involved dilute bleach baths; however, the results from various trials were inconclusive. Therefore, there is a need for more-targeted therapies that focus on restoring a healthy skin microbiome in AD patients, reducing the overgrowth of pathogenic triggers of AD, and promoting the recovery of beneficial commensal bacteria. These targeted therapies encompass various approaches, such as the use of probiotics, prebiotics, or synbiotics, reintroducing beneficial commensals to AD lesions, or even employing phage therapies. These options aim to provide more-effective and -sustainable solutions for managing AD while minimizing the risks associated with broad-spectrum antibiotics [129].

Probiotics and prebiotics complement each other when used to improve health. The combination of both concepts is called synbiotics, where prebiotics (or substrates) enhance the survival of probiotic strains and prolong the retention period of specific probiotics. Synbiotics can optimize, maintain, and restore the skin microbiota systemically (through ingestion) or through topical applications [153–156]. Applying probiotic bacteria topically directly impacts the targeted skin area, reinforcing the body's natural defense mechanisms. Probiotics, similar to the body's native bacteria, can generate antimicrobial peptides that support skin immune responses and combat pathogens. Nutritional products containing prebiotics and/or probiotics yield favorable effects on the skin by regulating the immune system and offering therapeutic advantages for conditions such as atopic diseases [153].

Probiotics modulate the overall microbiome and the immune system, potentially responsible for allergic reactions and the severity of atopic dermatitis. They inhibit the Th2-cell-mediated response, reducing pro-inflammatory cytokines, improving the Th1/Th2 ratio, decreasing INF-$\gamma$, stimulating phagocytosis, and increasing IgA levels [152]. Probiotics also exert their effects on the surface of epithelial cells by influencing the strength of the epithelial cell barrier and controlling the function, protein expression, and mucin secretion [157]. Probiotic bacteria generate substantial quantities of short-chain fatty acids by fermenting dietary fiber, and these acids exhibit strong anti-inflammatory properties and support the health of epithelial tissues. Probiotics can also regulate both innate and adaptive immune responses [151]. The effectiveness relies on the assumption that these positive outcomes take place throughout the body, as probiotics produce targeted effects in the intestinal environment, as well as on epithelial and immune cells, which may possess anti-allergic properties [153].

The administration of *Lactobacillus* and other probiotics in pregnant women has been shown to halve the risk of developing AD in children at 2 years of age [158]. Other recent study describes that administering probiotics during pregnancy and early childhood seems to have a positive effect in preventing the development of atopic dermatitis (AD) in infants [159]. However, the effectiveness of probiotics as a treatment is still unclear, needing further studies.

Improving nutritional status, nutrient digestion, specific and nonspecific immune responses, and beneficial effects on the gastrointestinal tract and skin are arguments to support the use of prebiotics and probiotics in patients with atopic dermatitis (AD). However, there is still insufficient data in the literature to answer questions about the optimal dosage, the ideal timing to initiate treatment, and the necessary duration to demonstrate beneficial effects [152].

A direct consequence of improving nutritional status is the increase in short-chain fatty acids (SCFAs), which are a group of organic compounds that consist of fewer than

six carbon atoms. They are primarily produced during the fermentation of dietary fiber by beneficial bacteria in the gut, particularly in the colon. Three of the most-common SCFAs are acetate (C2), propionate (C3), and butyrate (C4). These compounds play various essential roles in the body, including their notable anti-inflammatory effects, which can be particularly relevant in the context of atopic dermatitis, particularly for the following reasons: (i) Regulation of immune cells: SCFAs have been shown to modulate the function of immune cells, such as T cells and regulatory T cells (Tregs). Tregs play a crucial role in dampening excessive immune responses and inflammation. SCFAs can promote the development and activity of Tregs, thereby helping to suppress inflammation. (ii) Inhibition of pro-inflammatory cytokines: SCFAs can inhibit the production of pro-inflammatory cytokines, such as tumor necrosis factor-alpha (TNF-alpha) and interleukin-6 (IL-6). Elevated levels of these cytokines are often observed in inflammatory skin conditions such as atopic dermatitis, and their reduction can help alleviate inflammation. (iii) Barrier function improvement: Atopic dermatitis is characterized by a compromised skin barrier, which allows allergens and irritants to penetrate the skin more easily, leading to inflammation. SCFAs can enhance the integrity of the skin barrier by promoting the production of skin proteins such as filaggrin and involucrin, which play crucial roles in maintaining skin barrier function. (iv) Anti-microbial effects: SCFAs possess antimicrobial properties that can help regulate the skin microbiome. By inhibiting the growth of harmful bacteria and promoting the growth of beneficial ones, SCFAs contribute to a healthier skin microbiome, which is important for reducing inflammation in conditions such as atopic dermatitis. (v) Reduction of oxidative stress: SCFAs have antioxidant properties, which means they can help neutralize harmful reactive oxygen species (ROSs) and reduce oxidative stress in the skin. Oxidative stress is known to exacerbate inflammation in various skin disorders, including atopic dermatitis. (vi) Improvement of mucosal immunity: SCFAs play a role in maintaining the health of mucosal surfaces, including the gut and respiratory tract. A strong mucosal immune system can help regulate systemic inflammation, which may indirectly benefit skin conditions such as atopic dermatitis [160].

Phage therapy is also being studied as microbiome-based therapeutics to treat AD microbiota dysbiosis. The use of staphylococcal phage SaGU1 was tested in combination with commensal bacteria *S. epidermidis* with good results using a mouse model [161]. The use of a recombinant phage endolysin, Staphefekt SA.100, is also being tested; however, the results are not yet conclusive [162,163]. In Table 3, therapies targeting the microbiota' composition in AD are exemplified.

**Table 3.** Therapies targeting microbiota' composition in AD.

| Therapy | Examples | References |
|---|---|---|
| Probiotics | *Lactobacillus, Bifidobaterium* | [152,158,159] |
| Prebiotics | Inulin, resistant starch, polydextrose, pectic oligosaccharides derived from pectin (fructooligosaccharides (FOS) and galactooligosaccharides (GOS)) | [153,164–166] |
| Synbiotics | *Bifidobacterium* and galactooligosaccharide | [167–170] |
| Improving nutritional status | Fibers, fruit, and Mediterranean diet consumption | [152] |
| Short-chain fatty acids (SCFAs) | Acetate (C2), propionate (C3), and butyrate (C4) | [160] |
| Phage therapy | Staphylococcal phage, SaGU1; phage endolysin, Staphefekt SA.100 | [161–163,171] |

To summarize, the results indicate that microbiome-based therapeutics offer several promising treatment avenues for AD. However, further investigation is needed to determine the specific strains of commensal bacteria that can yield long-lasting effects [171,172].

## 7. Conclusions

The microbiome has significant relevance to health and is, therefore, considered an important therapeutic target. Characterizing microbiome locations such as the gastrointestinal tract, skin, and urogenital tract is essential for understanding its connection to pathological conditions. The balance among microbial species present in the human body is crucial for maintaining organismal homeostasis, making it extremely important for human health. When there is a disruption in this balance, it can lead to pathological conditions such as obesity, gastrointestinal diseases, inflammatory disorders, cardiovascular diseases, and skin diseases, namely AD, among others [173].

The role of the gut and skin microbiome is more complex and intriguing than previously thought. Both the skin and intestinal microbiomes influence the development and function of the immune tissue. The microbiome has been identified as a potential trigger for immune system dysregulation. Therefore, manipulation of the microbiota for therapeutic purposes can be achieved through the use of prebiotics or probiotics. Modulating the microbiome may be effective in the treatment of inflammatory skin diseases. Promising future therapeutic strategies will aim to prevent microbial dysbiosis. Nevertheless, additional research is required to determine if it is possible to synchronize the existing treatment methods for AD with the skin microbiota to enhance clinical outcomes. Additionally, investigating whether directly modifying the microbiota as a novel therapeutic approach can be made as effective as possible without introducing any therapeutic complications is essential [100].

Microbiome research holds tremendous potential for interventions in diagnosis and therapy, once it is characterized by its plasticity, allowing for beneficial modulation, thereby improving human health and potentially preventing and reducing the development of various diseases. By modulating the microbiome through the use of probiotics, prebiotics, dietary interventions, and fecal bacterial transplantation, it becomes possible to prevent the onset and progression of certain pathologies. However, despite all these advances, much remains to be discovered. It is hoped that future research will bring new therapeutics resulting in an improvement in the quality of human life.

**Funding:** This research was funded by the project "Health from Portugal—HfPT" [C630926586-00465198], financed by PRR—Plano de Recuperação e Resiliência under the Next Generation EU from the European Union.

**Institutional Review Board Statement:** Not applicable.

**Informed Consent Statement:** Not applicable.

**Data Availability Statement:** Not applicable.

**Conflicts of Interest:** The author declares no conflict of interest.

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
