# Peer review of "An Overview of the Relevance of Human Gut and Skin Microbiome in Disease: The Influence on Atopic Dermatitis"

_applsci, doi:10.3390/app131810540_

Round 1

Reviewer 1 Report

This manuscript addresses a timely topic and makes a relevant contribution to the field which elaborates manipulating of microbiota to treat Atopic Dermatitis. However, some major revisions are needed before it can be published.

1. The manuscript tries to achieve too many things at the same time. The authors need to narrow down their research focus. I encourage the authors to provide more in-depth evidence on focused points.

2. The author only describes several points but fails to justify any with proper molecular mechanism.

The author must focus on the skin Microbiota alternation in AD Patients by providing summary of the recent studies demonstrating the dysbiosis of skin microbiota in AD.

3.  It could have been better if the author can provide a flowchart for the mechanisms of how skin microbiota regulates AD pathogenesis.

4. The author must include the gut microbiota profiles in AD patient by summarizing of clinical studies demonstrating the alternation of gut microbiota in patients with AD.

5. How the gut microbiota regulates AD-related immune responses and the underlying mechanisms? Please provide a figure by defining mechanisms of how gut microbiota regulate AD pathogenesis.

6. Under “Therapies targeting microbiota composition” section, the author should include the role of short-chain fatty acids and their anti-inflammatory effects in AD.

7. Throughout the manuscript, there are several language mistakes. Therefore, I recommend the paper should undergo professional language editing before it can be published.

8. The conclusion is too long and should be shortened.

9. There are several repetitive and overlapping contents in introduction and conclusion section which could have been avoided.

10. There are several typos which could have been corrected before the final submission.

Throughout the manuscript, there are several language mistakes. Therefore, I recommend the paper should undergo professional language editing before it can be published.

Author Response

Reviewer 1

Answers for reviewer 1:

Dear reviewer

Thank you very much for your helpful and important comments which were taken into consideration and responded to below and marked on the manuscript in blue.

Comment:

This manuscript addresses a timely topic and makes a relevant contribution to the field which elaborates manipulating of microbiota to treat Atopic Dermatitis. However, some major revisions are needed before it can be published.

Answer: Thanks for your important comments. Detailed answers are given for each point.

Comment 1: The manuscript tries to achieve too many things at the same time. The authors need to narrow down their research focus. I encourage the authors to provide more in-depth evidence on focused points.

Answer: Changes were made throughout the entire document taking this comment into consideration, particularly in section 5. Also, a connection between sections was performed. Changed are marked in blue.

Comment 2: The author only describes several points but fails to justify any with proper molecular mechanism.The author must focus on the skin Microbiota alternation in AD Patients by providing a summary of the recent studies demonstrating the dysbiosis of skin microbiota in AD.

Answer: Section 5.2 section was added detailing the required information, page 13.

Comment 3: It could have been better if the author can provide a flowchart for the mechanisms of how skin microbiota regulates AD pathogenesis

Answer: Figure 2 was added, page 14.

Comment 4: The author must include the gut microbiota profiles in AD patient by summarizing of clinical studies demonstrating the alternation of gut microbiota in patients with AD.

Answer: Section 5.5 was added, page 15.

Comment 5: How the gut microbiota regulates AD-related immune responses and the underlying mechanisms? Please provide a figure by defining mechanisms of how gut microbiota regulate AD pathogenesis.

Answer: Figure 3 was added as long as a brief explanation in Section 5.5 was added, page 16.

Comment 6: Under “Therapies targeting microbiota composition” section, the author should include the role of short-chain fatty acids and their anti-inflammatory effects in AD.

Answer: The required information was added, page 19.

Comment 7: Throughout the manuscript, there are several language mistakes. Therefore, I recommend the paper should undergo professional language editing before it can be published.

Answer: The revised manuscript has undergone professional language editing correcting language mistakes as requested. As mentioned, the entire document was edited, however, the main changes are marked in yellow.

Comment 8: The conclusion is too long and should be shortened.

Answer: The conclusion has been shortened, page 20.

Comment 9: There are several repetitive and overlapping contents in introduction and conclusion section which could have been avoided.

Answer: Answering comment 8, the authors have also taken into consideration comment 9 therefore overlapping was removed.

Comment 10: There are several typos which could have been corrected before the final submission.

Answer: Solving comment 7 this comment was also solved.

Comments on the Quality of English Language

Throughout the manuscript, there are several language mistakes. Therefore, I recommend the paper should undergo professional language editing before it can be published.

Answer: Same as comments 7 and 10.

Reviewer 2 Report

This manuscript deals with an outstanding topic related to the human microbiome. However, several issues need to be solved before its publication in the Applied Sciences Journal.

I've included my detailed comments and suggestions in the attached file. Please revise them, and I will be honored to review the revised version of this manuscript.

This manuscript's English quality is fair, and minor spelling mistakes must be corrected.

Author Response

Reviewer 2

Answers for reviewer 2:

Dear reviewer

Thank you very much for your helpful and important comments which were taken into consideration and responded to below and marked on the manuscript in green.

Comment:

This manuscript deals with an outstanding topic related to the human microbiome. However, several issues need to be solved before its publication in the Applied Sciences Journal.

I've included my detailed comments and suggestions in the attached file. Please revise them, and I will be honored to review the revised version of this manuscript.

Answer: Thanks for your important comments. Detailed answers are given for each point.

Comment: Page 1 Title (in the sent PDF); human gut microbiome since the author says this review has the objective to focus on gut microbiota

Answer: The title was changed according to the objectives, page 1.

Comment: Page 1 Abstract (in the sent PDF); The authors should replace this sentence in terms of their own problem "skin microbiota and atopic dermatitis”

Answer: The paragraph was changed according to the objectives, page 1.

Comment: Page 1 Keywords (in the sent PDF): can the authors suggest on genus or species particularly related with skin microbiota?, please clarify

Answer: The author add the most important species regarding skin dysbiosis, page 1.

Comment: Page 1 Line 33-37 (in the sent PDF); these sentences need a reference

Answer: A reference was added, page 1.

Comment: Page 1 Line 40 (in the sent PDF); which ones?, please add  at least three examples

Answer: Examples were added as requested, page 1.

Comment: Page 1 Line 44 (in the sent PDF); also, lifestyle and exercise have a powerful impact in the composition of human microbiome...

Answer: The suggestion was added to the manuscript, page 1/2.

Comment: Page 2 Line 54 (in the sent PDF); I disagree with this sentence, gut microbiota modulates other axis and diseases beyond cardiovascular diseases, please clarify.

Answer: The sentence was changed, page 2.

Comment: Page 2 Line 56 (in the sent PDF); gut

Answer: Change was done, page 2.

Comment: Page 2 Line 58 (in the sent PDF); which microorganisms, please clarify

Answer: Microorganisms were specified, page 2.

Comment: Page 2 Line 65 (in the sent PDF); 5this section is a quite confusing when we compare with the previous paragraphs, please clarify. Also is not clear, what was the methodological survey to select the articles to discuss in this review, please clarify

Answer: The section was changed and adjusted to the previous paragraphs ta have also been changed. The methodology survey was detailed but the manuscript's aim is not a systematic review, but an overview of relevant issues concerning microbiome relation with disease in particular discussing the case of AD page 2.

Comment: Page 2 Line 72(in the sent PDF); needs a reference

Answer: Reference was added, page 2.

Comment: Page 2 Line 79 (in the sent PDF); the same, this is great information, but actually needs a reference.

Answer: References were added, page 2.

Comment: Page 2 Line 91 (in the sent PDF); which ones?.

Answer: The physiological processes were detailed (underlined), “The human microbiome provides a fundamental internal ecosystem for numerous physiological processes, among which some can be highlighted, such as protection against pathogens, nutrient processing, stimulation of angiogenesis, development and maintenance of the intestinal epithelial barrier, and maintenance of the immune system” page 2.

Comment: Page 3 Line 106 (in the sent PDF); this section is disconnected with the previous one, please clarify, also needs a reference.

Answer: Changes were done according to the suggestion, page 3.

Comment: Page 3 Line 112 (in the sent PDF); needs a reference

Answer: Reference was added, page 3.

Comment: Page 3 Line 132 (in the sent PDF); which microorganisms please clarify

Answer: Microorganisms were specified and references were added, page 3.

Comment: Page 3 Line 140 (in the sent PDF); adult or children, male or female, which population, age?, please clarify this section is so general and practically did not say anything relevant

Answer: The paragraph was changed, page 3.

Comment: Page 3 Line 145 (in the sent PDF); which acids?

Answer: Acid was specified, page 4.

Comment: Page 4 Line 179 (in the sent PDF); needs a reference

Answer: Reference was added, page 4.

Comment: Page 5 Line 210 (in the sent PDF); is a good phrase but needs a reference

Answer: References were added, page 5.

Comment: Page 5 Line 253 (in the sent PDF); needs a reference

Answer: Reference was added, page 6.

Comment: Page 7 Line 336 (in the sent PDF); needs a reference

Answer: Reference was added, page 7.

Comment: Page 7 Line 336 (in the sent PDF); taxa names must be in italics

Answer: According to the rules of microbial taxonomy, only the genus and species should be in italics. All other taxonomic groups should not be written in italics. However, if the reviewer and the editor deem it appropriate, they can be changed, page 7.

Comment: Page 8 Table 1 (in the sent PDF); this table is great, but how the author get this information, please clarify

Answer: References were added, page 8.

Comment: Page 8 Line 364 (in the sent PDF); needs a reference

Answer: Reference was added, page 8.

Comment: Page 10 Line 447 (in the sent PDF); needs a reference

Answer: Reference was added, page 8.

Comment: Page 10 Line 480 (in the sent PDF); this topic is so important I think that the author needs to improve and strenght this section

Answer: The section was improved, page 10.

Comment: Page 11 Line 507 (in the sent PDF); until this point is not clear what was the relationship of all microbiomes with atopic dermatitis, authors need to improve the conectiveness between sections

Answer: Text was changed, page 11.

Comment: Page 11 Line 538 (in the sent PDF); microbiota, flora refers to "plants", author need to change all "flora or microflora" terms.

Answer: Suggestions have been followed throughout the entire document.

Comment: Page 13 Line 606 (in the sent PDF); skin microbiota, gut, urogenital?, please clarify.

Answer: Gut was added, page 17.

Comment: Page 13 Line 614 (in the sent PDF); these therapies are related to any particular disease?, please clarify.

Answer: These therapies are for general health status, A sentence was added, page 17.

Comment: Page 14 Line 626; needs a reference

Answer: Reference was added, page 18.

Comment: Page 16 Line 695; needs a reference

Answer: References were added, page 20.

Round 2

Reviewer 1 Report

The author have satisfactorily addressed most of my concerns. 

The author have satisfactorily addressed most of my concerns. Though the revised version is much better than the first one but there is further scope for improvement. Editor may take a call.

Author Response

Reviewer 1

Answers for reviewer 1:

Dear reviewer

Thank you very much for your comments.

Comment:

The author have satisfactorily addressed most of my concerns.

Answer: Thanks for your comments.

Comments on the Quality of English Language

The author have satisfactorily addressed most of my concerns. Though the revised version is much better than the first one but there is further scope for improvement. Editor may take a call.

Answer: The author will make any suggestions needed, when asked.

Reviewer 2 Report

The authors have addressed all my comments and suggestions. They need to improve figures 2 and 3. They could use any related tool such as Biorender or Canva.

Minor editing of English language required

Author Response

Reviewer 2

Answers for reviewer 2:

Dear reviewer

Thank you very much for your comments which were taken into.

Comment:

The authors have addressed all my comments and suggestions. They need to improve figures 2 and 3. They could use any related tool such as Biorender or Canva.

Answer: Thanks for your comments. Figures 2 and 3 were improved. The author made the figures simpler and clearer. No artifacts were added since reviewer 2 in the first review requested something schematic like a flowchart. Figures 2 and 3 will also be attached as separate documents so that they can have more definition/quality if necessary.
